# Neural End-to-End Learning for Computational Argumentation Mining

## Abstract

We investigate neural techniques for end-to-end computational argumentation mining. We frame the problem as a token-based dependency parsing as well as a token-based sequence tagging model, including a multi-task learning setup. Contrary to models that operate on the argument component level, we find that framing the problem as dependency parsing leads to subpar performance results. In contrast, less complex (local) tagging models based on BiLSTMs perform robustly across classification scenarios, being able to catch long-range dependencies inherent to the argumentation mining problem. Moreover, we find that jointly learning 'natural' subtasks, in a multi-task learning setup, improves performance.

## 1 Introduction

Computational argumentation mining (AM) deals with finding argumentation structures in text. This involves several subtasks, such as: (a) separating argumentative units from non-argumentative units, also called 'component segmentation'; (b) classifying argument components into classes such as "Premise" or "Claim"; (c) finding relations between argument components; (d) classifying relations into classes such as "Support" or "Attack" (Persing and Ng, 2016; Stab and Gurevych, 2016).

Thus, AM would have to detect claims and premises (reasons) in texts such as the following, where premise P supports claim C:

> Since    it killed many marine lives$_P$    ,
> tourism has threatened nature$_C$ .

Argument structures in real texts are typically much more complex, cf. Figure 1.

While different research has addressed different subsets of the AM problem (see below), the ultimate goal is to solve all of them, starting from unannotated plain text. Two recent approaches to this end-to-end learning scenario are Persing and Ng (2016) and Stab and Gurevych (2016). Both solve the end-to-end task by first training independent models for each subtask and then defining an integer linear programming (ILP) model that encodes global constraints such as that each premise has a parent, etc. Besides their pipeline architecture the approaches also have in common that they heavily rely on hand-crafted features.

Hand-crafted features pose a problem because AM is to some degree an "arbitrary" problem in that the notion of "argument" critically relies on the underlying argumentation theory (Reed et al., 2008; Biran and Rambow, 2011; Habernal and Gurevych, 2015; Stab and Gurevych, 2016). Accordingly, datasets typically differ with respect to their annotation of (often rather complex) argument structure. Thus, feature sets would have to be manually adapted to and designed for each new sample of data, a challenging task. The same critique applies to the designing of ILP constraints. Moreover, from a machine learning perspective, pipeline approaches are problematic because they solve subtasks independently and thus lead to error propagation rather than exploiting interrelationships between variables. In contrast to this, we investigate *neural* techniques for end-to-end learning in computational AM, which do not require the hand-crafting of features or constraints. The models we survey also all capture some notion of "joint"—rather than "pipeline"—learning. We investigate several approaches.

First, we frame the end-to-end AM problem as a dependency parsing problem. Dependency parsing may be considered a natural choice for AM, because argument structures often form trees, or

closely resemble them (see §3). Hence, it is not surprising that "discourse parsing" has been suggested for AM (Peldszus and Stede, 2015). What distinguishes our approach from these previous ones is that we operate on the *token* level, rather than on the level of components, because we address the end-to-end framework and, thus, do not assume that non-argumentative units have already been sorted out and/or that the boundaries of argumentative units are given.

Second, we frame the problem as a sequence tagging problem. This is a "natural" choice especially for component segmentation and typing, which is a typical entity recognition problem for which BIO tagging is a standard approach, pursued in AM, e.g., by Habernal and Gurevych (2016). The challenge in the end-to-end setting is to also include relations into the tagging scheme, which we realize by coding the distances between linked components into the tag label. Since related entities in AM are oftentimes several dozens of tokens apart from each other, neural sequence tagging models are in principle ideal candidates for such a framing because they can take into account *long-range dependencies*—something that is inherently difficult to capture with traditional feature-based tagging models such as conditional random fields (CRFs).

Third, we frame AM as a *multi-task* (tagging) problem (Caruana, 1997; Collobert and Weston, 2008). We experiment with subtasks of AM— e.g., component detection—as auxiliary tasks and investigate whether this improves performance on the AM problem. Adding such subtasks can be seen as analogous to de-coupling, e.g., component detection from the full AM problem.

Fourth, we evaluate the model of Miwa and Bansal (2016) that combines sequential (entity) and tree structure (relation) information and is in principle applicable to any problem where the aim is to extract entities and their relations. As such, this model makes fewer assumptions than our dependency parsing and tagging approaches.

The contributions of this paper are as follows. (1) We present the *first neural end-to-end* solutions to computational AM. (2) We show that several of them perform better than the state-of-the-art joint ILP model. (3) We show that a framing of AM as a token-based dependency parsing problem is ineffective—in contrast to what has been proposed for systems that operate on the coarser

component level and that (4) a standard neural sequence tagging model that encodes distance information between components performs robustly in different environments. Finally, (5) we show that a multi-task learning setup where natural subtasks of the full AM problem are added as auxiliary tasks improves performance.

## 2 Related Work

AM has applications in legal decision making (Palau and Moens, 2009; Moens et al., 2007), document summarization, and the analysis of scientific papers (Kirschner et al., 2015). Its importance for the educational domain has been highlighted by recent work on writing assistance (Zhang and Litman, 2016) and essay scoring (Persing and Ng, 2015; Somasundaran et al., 2016).

Most works on AM address subtasks of AM such as locating/typing components (Florou et al., 2013; Moens et al., 2007; Rooney et al., 2012; Knight et al., 2003; Levy et al., 2014; Rinott et al., 2015). Relatively few works address the full AM problem of component *and* relation identification. Peldszus and Stede (2016) present a corpus of microtexts containing only argumentatively relevant text of controlled complexity. To our best knowledge, Stab and Gurevych (2016) created the only corpus of attested high quality which annotates the AM problem in its entire complexity: it contains token-level annotations of components, their types, as well as relations and their types.

## 3 Data

We use the dataset of persuasive essays (PE) from Stab and Gurevych (2016), which contains student essays written in response to controversial topics such as "competition or cooperation—which is better?"

The corpus consists of 402 essays, 80 of which are reserved for testing. The annotation distinguishes between **major claims** (the central position of an author with respect to the essay's topic), **claims** (controversial statements that are either *for* or *against* the major claims), and **premises**, which give reasons for claims or other premises and either *support* or *attack* them. Overall, there are 751 major claims, 1506 claims, and 3832 premises.

The corpus has a special structure, illustrated in Figure 1. First, major claims relate to no other components. Second, claims always relate to all other major claims. Third, each premise relates

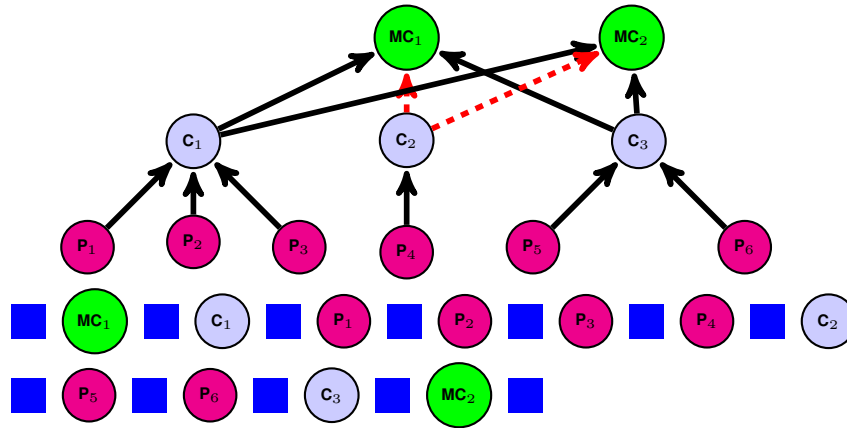

Figure 1: Bottom: Linear argumentation structure in a student essay. The essay is comprised of non-argumentative units (square blue) and argumentative units of different types: Premises (P), claims (C) and major claims (MC). Top: Relationsships between argumentative units. Black arrows are support (for), red dashed arrows are attack (against).

to exactly one claim or premise. Thus, the argument structure in each essay is—almost—a tree. Since there may be several major claims, each claim connects to multiple targets, which violates the tree structure. This does not pose a problem, however, since we can "loss-lessly" re-link the claims to one of the major claims (e.g., the last major claim in a document) and create a special root node to which the major claims link.

There is another peculiarity of this data. Each essay is divided into paragraphs, of which there are 2235 in total (1786 train, 449 test). The argumentation structure is completely contained within a paragraph, except, possibly, for the relation from claims to major claims. Paragraphs have an average length of 66 tokens and are therefore much shorter than essays, which have an average length of 368 tokens. Thus, prediction on the paragraph level is easier than prediction on the essay level, because there are fewer components in a paragraph and hence fewer possibilities of source and target components in argument relations. Paragraphs are not only easier for relation classification, but also for component classification: a paragraph can never contain premises only, for example, since premises link to other components.

## 4 Models

This section describes our neural network framings for end-to-end AM.

**Sequence Tagging** is the problem of assigning each element in a stream of input tokens a label. In a neural context, the natural choice for tagging problems are recurrent neural nets (RNNs) in which a hidden vector representation $\mathbf{h}_t$ at time point $t$ depends on the previous hidden vector representation $\mathbf{h}_{t-1}$ and the input $\mathbf{x}_t$. In this way, an infinite window ("long-range dependencies") around the current input token $\mathbf{x}_t$ can be taken into account when making an output prediction $\mathbf{y}_t$. We choose particular RNNs, namely, LSTMs (Hochreiter and Schmidhuber, 1997), which are popular for being able to address vanishing/exploding gradients problems. In addition to considering a left-to-right flow of information, bidirectional LSTMs also capture information to the right of the current input token.

The most recent generation of neural tagging models add label dependencies to BiLSTMs (BL), so that successive output decisions are not made independently. This class of models is called BiLSTM-CRF (BLC) (Huang et al., 2015). The model of Ma and Hovy (2016) adds convolutional neural nets (CNNs) on the character-level to BiLSTM-CRFs, leading to BiLSTM-CRF-CNN (BLCC) models. The character-level CNN may address problems of out-of-vocabulary words, that is, words not seen during training.

**AM as Sequence Tagging**: We frame AM as the following sequence tagging problem. Each input token has an associated label from $\mathcal{Y}$, where

$$\mathcal{Y} = \{(b, t, d, s) \mid b \in \{\text{B}, \text{I}, \text{O}\}, t \in \{\text{P}, \text{C}, \text{MC}, \bot\},$$
$$d \in \{\dots, -2, -1, 1, 2, \dots, \bot\}, \quad (1)$$
$$s \in \{\text{Supp}, \text{Att}, \text{For}, \text{Ag}, \bot\}\}.$$

In other words, $\mathcal{Y}$ consists of all four-tuples $(b, t, d, s)$ where $b$ is a BIO encoding indicating whether the current token is non-argumentative (O) or begins (B) or continues (I) a component; $t$ indicates the *type* of the component (claim C, premise P, or major claim MC for our data). Moreover, $d$ encodes the distance—measured in number of components—between the current component and the component it relates to. We encode the same $d$ value for each token in a given component. Finally, $s$ is the relation type ("stance") between two components and its value may be Support (Supp), Attack (Att), or For or Against (Ag). We also have a special symbol $\perp$ that indicates when a particular slot is not filled: e.g., a non-argumentative unit ($b = $ O) has neither component type, nor relation, nor relation type. We refer to this framing as $\mathtt{STag}_T$ (for "*Simple Tagging*"), where $T$ refers to the tagger used. For the example from §1, our coding would hence be:

| | | | |
|---|---|---|---|
| Since | it | killed | many |
| (O,$\perp$,$\perp$,$\perp$) | (B,P,1,Supp) | (I,P,1,Supp) | (I,P,1,Supp) |
| marine | lives | , | tourism |
| (I,P,1,Supp) | (I,P,1,Supp) | (O,$\perp$,$\perp$,$\perp$) | (B,C,$\perp$,For) |
| has | threatened | nature | . |
| (I,C,$\perp$,For) | (I,C,$\perp$,For) | (I,C,$\perp$,For) | (O,$\perp$, $\perp$, $\perp$) |

While the size of the label set $\mathcal{Y}$ is potentially infinite, we would expect it to be finite even in a potentially infinitely large data set, because humans also have only finite memory and are therefore expected to keep related components close in textual space. Indeed, as Figure 2 shows, in our PE essay data set about 30% of all relations between components have distance $-1$, that is, they follow the claim or premise that they attach to. Overall, around 2/3 of all relation distances $d$ lie in $\{-2, -1, 1\}$. However, the figure also illustrates that there are indeed long-range dependencies: distance values between $-11$ and $+10$ are observed in the data.

**Dependency Parsing** methods can be classified into graph-based and transition-based approaches (Kiperwasser and Goldberg, 2016). *Transition-based* parsers encode the parsing problem as a sequence of configurations which may be modified by application of actions such as shift, reduce, etc. The system starts with an initial configuration in which sentence elements are on a *buffer* and a *stack*, and a classifier successively decides which action to take next, leading to different configura-

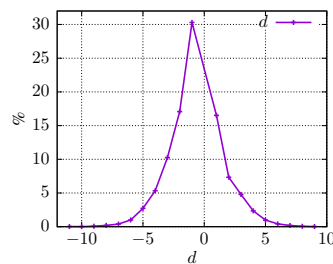

Figure 2: Distribution of distances $d$ between components in PE dataset.

tions. The system terminates after a finite number of actions, and the parse tree is read off the terminal configuration. *Graph-based* parsers solve a structured prediction problem in which the goal is learning a scoring function over dependency trees such that correct trees are scored above all others.

Traditional dependency parsers used handcrafted feature functions that look at "core" elements such as "word on top of the stack", "POS of word on top of the stack", and conjunctions of core features such as "word is X and POS is Y" (see McDonald et al. (2005)). Most neural parsers have not entirely abandoned feature engineering. Instead, they rely, for example, on encoding the core features of parsers as low-dimensional embedding vectors (Chen and Manning, 2014) but ignore feature combinations. Kiperwasser and Goldberg (2016) design a neural parser that uses only four features: the BiLSTM vector representations of the top 3 items on the stack and the first item on the buffer. In contrast, Dyer et al. (2015)'s neural parser associates each stack with a "stack LSTM" that encodes their contents. Actions are chosen based on the stack LSTM representations of the stacks, and no more feature engineering is necessary. Moreover, their parser has thus access to any part of the input, its history and stack contents.

**AM as Dependency Parsing**: To frame a problem as a dependency parsing problem, each instance of the problem must be encoded as a directed tree, where tokens have heads, which in turn are labeled. For end-to-end AM, we propose the framing illustrated in Figure 3. We highlight two design decisions, the remaining are analogous and/or can be read off the figure.

- The head of each non-argumentative text token is the document terminating token END, which is a punctuation mark in all our cases. The label of this link is O, the symbol for

non-argumentative units.

- The head of each token in a premise is the *first* token of the claim or premise that it links to. The label of each of these links is $(b, \text{P}, \text{Supp})$ or $(b, \text{P}, \text{Att})$ depending on whether a premise "supports" or "attacks" a claim or premise; $b \in \{\text{B}, \text{I}\}$.

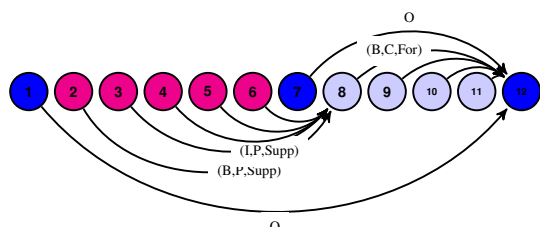

Figure 3: Dependency representation of sample sentence from §1. Links and selected labels.

**Multi-Task Learning** Recently, there has been a lot of interest in so-called multi-task learning (MTL) scenarios, where several tasks are learned *jointly* (Søgaard and Goldberg, 2016; Peng and Dredze, 2016; Yang et al., 2016; Rusu et al., 2016; Héctor and Plank, 2017). It has been argued that such learning scenarios are closer to human learning because humans often transfer knowledge between several domains. In a neural context, MTL is often implemented via weight sharing: several tasks are trained in the same network architecture, thereby sharing a substantial portion of network's parameters. This forces the network to learn *generalized* representations.

In the MTL framework of Søgaard and Goldberg (2016) the underlying model is a BiLSTM with several hidden layers. Then, given different tasks, each task $k$ 'feeds' from one of the hidden layers in the network. In particular, the hidden states encoded in a specific layer are fed into a multiclass classifier $f_k$. The same work has demonstrated that this MTL protocol may be successful when there is a hierarchy between tasks and 'lower' tasks feed from lower layers.

**AM as MTL**: We use the same framework $\text{STag}_T$ for modeling AM as MTL. However, we in addition train auxiliary tasks in the network—each with a distinct label set $\mathcal{Y}'$.

**LSTM-ER** Miwa and Bansal (2016) present a neural end-to-end system for identifying both entities as well as relations between them. Their entity

dection system is a BLC tagger and their relation detection system is a neural net that predicts a relation for each pair of detected entities. This relation module is a TreeLSTM model that makes use of dependency tree information. In addition to decoupling entity and relation detection but jointly modeling them, pretraining on entities and scheduled sampling (Bengio et al., 2015) is applied to prevent low performance at early training stages of entity detection and relation classification. To adapt LSTM-ER for the argumentative structure encoded in the PE dataset, we model three types of entities (premise, claim, major claim) and four types of relations (for, against, support, attack).

We use the **feature-based ILP model** from Stab and Gurevych (2016) as a comparison system. This system solves the subtasks of AM—component segmentation, component typing, relation identification and typing—independently. Afterwards, it defines an ILP model with various constraints to enforce valid argumentation structure. As features it uses structural, lexical, syntactic and context features, cf. Stab and Gurevych (2016) and Persing and Ng (2016).

Summarizing, we distinguish our framings in terms of *modularity* and in terms of their *constraints*. *Modularity*: Our dependency parsing framing and LSTM-ER are more modular than $\text{STag}_T$ because they de-couple relation information from entity information. However, part of this modularity can be regained by using $\text{STag}_T$ in an MTL setting. Moreover, since entity and relation information are considerably different, such a de-coupling may be advantageous. *Constraints*: LSTM-ER can, in principle, model any kind of—even many-to-many—relationships between detected entities. Thus, it is not guaranteed to produce trees, as we observe in AM datasets. $\text{STag}_T$ also does not need to produce trees, but it more severely restricts search space than does LSTM-ER: each token/component can only relate to one (and not several) other tokens/components. The same constraint is enforced by the dependency parsing framing. All of the tagging modelings, including LSTM-ER, are *local* models whereas our parsing framing is a *global* model: it globally enforces a tree structure on the token-level.

Further remarks: (1) part of the TreeLSTM modeling inherent to LSTM-ER is ineffective for our data because this modeling exploits dependency tree structures on the *sentence* level,

while relationships between components are almost never on the sentence level. In our data, roughly 92% of all relationships are between components that appear in different sentences. Secondly, (2) that a model *enforces* a constraint does not necessarily mean that it is more suitable for a respective task. It has frequently been observed that models tend to produce output consistent with constraints in their training data in such situations (Zhang et al., 2017; Héctor and Plank, 2017); thus, they have *learned* the constraints.

## 5 Experiments

This section presents and discusses the empirical results for the AM framings outlined in §4. We relegate issues of *pre-trained word embeddings*, *hyperparameter optimization* and further *practical issues* to the supplementary material. Links to software used as well as some additional error analysis can also be found there.

**Evaluation Metric** We adopt the evaluation metric suggested in Persing and Ng (2016). This computes true positives TP, false positives FP, and false negatives FN, and from these calculates component and relation $F_1$ scores as $F_1 = \frac{2TP}{2TP+FP+FN}$. For space reasons, we refer to Persing and Ng (2016) for specifics, but to illustrate, for *components*, true positives are defined as the set of components in the gold standard for which there exists a predicted component with the same type that 'matches'. Persing and Ng (2016) define a notion of what we may term 'level $\alpha$ matching': for example, at the 100% level (exact match) predicted and gold components must have exactly the same spans, whereas at the 50% level they must only share at least 50% of their tokens (approximate match). We refer to these scores as C-F1 (100%) and C-F1 (50%), respectively. For *relations*, an analogous F1 score is determined, which we denote by R-F1 (100%) and R-F1 (50%). We note that R-F1 scores depend on C-F1 scores because correct relations must have correct arguments. We also define a 'global' F1 score, which is the F1-score of C-F1 and R-F1.

Most of our results are shown in Table 1.

**(a) Dependency Parsing** We show results for the two feature-based parsers MST (McDonald et al., 2005), Mate (Bohnet and Nivre, 2012) as well as the neural parsers by Dyer et al. (2015) (LSTM-Parser) and Kiperwasser and Goldberg

(2016) (Kiperwasser). We train and test all parsers on the paragraph level, because training them on essay level was typically too memory-exhaustive.

MST mostly labels only non-argumentative units correctly, except for recognizing individual major claims, but never finds their exact spans (e.g., *"tourism can create negative impacts on"* while the gold major claim is *"international tourism can create negative impacts on the destination countries"*). Mate is slightly better and in particular recognizes several major claims correctly. Kiperwasser performs decently on the approximate match level, but not on exact level. Upon inspection, we find that the parser often predicts 'too large' component spans, e.g., by including following punctuation. The best parser by far is the LSTM-Parser. It is over 100% better than Kiperwasser on exact spans and still several percentage points on approximate spans.

How does performance change when we switch to the essay level? For the LSTM-Parser, the best performance on essay level is 32.84%/47.44% C-F1 (100%/50% level), and 9.11%/14.45% on R-F1, but performance result varied drastically between different parametrizations. Thus, the performance drop between paragraph and essay level is in any case immense.

Since the employed features of modern feature-based parsers are rather general—such as distance between words or word identities—we had expected them to perform much better. The minimal feature set employed by Kiperwasser is apparently not sufficient for accurate AM but still a lot more powerful than the hand-crafted feature approaches. We hypothesize that the LSTM-Parser's good performance, relative to the other parsers, is due to its encoding of the *whole* stack history—rather than just the top elements on the stack as in Kiperwasser— which makes it aware of much larger 'contexts'. While the drop in performance from paragraph to essay level is expected, the LSTM-Parser's deterioration is much more severe than the other models' surveyed below. We believe that this is due to the very long sequences encountered on essay level. Dependency parsers' global view upon the data may be too complex a modeling in such situations and/or would require much more training data: the number of possible trees to consider on $n$ tokens may just be too huge a search space when $n$ is large.

|  | Paragraph level | | | | | | | | Essay level | | | | | |
|---|---|---|---|---|---|---|---|---|---|---|---|---|---|---|
|  | Acc. | C-F1 | | R-F1 | | F1 | | | Acc. | C-F1 | | R-F1 | | F1 | |
|  |  | 100% | 50% | 100% | 50% | 100% | 50% | | | 100% | 50% | 100% | 50% | 100% | 50% |
| MST-Parser | 31.23 | 0 | 6.90 | 0 | 1.29 | 0 | 2.17 | | | | | | | | |
| Mate | 22.71 | 2.72 | 12.34 | 2.03 | 4.59 | 2.32 | 6.69 | | | | | | | | |
| Kiperwasser | 52.80 | 26.65 | 61.57 | 15.57 | 34.25 | 19.65 | 44.01 | | | | | | | | |
| LSTM-Parser | 55.68 | 58.86 | 68.20 | 35.63 | 40.87 | 44.38 | 51.11 | | | | | | | | |
| $\mathtt{STag_{BLCC}}$ | 59.34 | 66.69 | 74.08 | 39.83 | 44.02 | 49.87 | 55.22 | | **60.46** | 63.23 | 69.49 | **34.82** | **39.68** | **44.90** | **50.51** |
| LSTM-ER | **61.67** | **70.83** | **77.19** | **45.52** | **50.05** | **55.42** | **60.72** | | 54.17 | **66.21** | **73.02** | 29.56 | 32.72 | 40.87 | 45.19 |
| ILP | 60.32 | 62.61 | 73.35 | 34.74 | 44.29 | 44.68 | 55.23 | | | | | | | | |

Table 1: Performance of dependency parsers, $\mathtt{STag_{BLCC}}$, LSTM-ER and ILP (from top to bottom). The ILP model operates on both levels. Best scores in each column in bold (significant at $p < 0.01$; Two-sided Wilcoxon signed rank test, pairing F1 scores for documents).

**(b) Sequence Tagging** For these experiments, we use the BLCC tagger from Ma and Hovy (2016) and refer to the resulting system as $\mathtt{STag_{BLCC}}$. Again, we observe that paragraph level is considerably easier than essay level. The major benefit is again seen for relations (about 5% points increase from essay to paragraph level). Overall, $\mathtt{STag_{BLCC}}$ is ∼13% better than the best parser for C-F1 and ∼11% better for R-F1 on the paragraph level. Our explanation is that taggers are simpler local models, and thus need less training data and are less prone to overfitting. Moreover, they can much better deal with the long sequences because they are largely invariant to length: every subsequence of a sequence forms a valid input to them, while the same is not true for parsers—a subsequence of a sequence does typically not form a tree.

**(c) MTL** As indicated, we use the MTL tagging framework from Søgaard and Goldberg (2016) for multi-task experiments. The underlying tagging framework is weaker than that of BLCC: there is no CNN which can take subword information into account and there are no dependencies between output labels: each tagging prediction is made independently of the other predictions. We refer to this system as $\mathtt{STag_{BL}}$.

Accordingly, as Table 2 shows for the essay level (paragraph level omitted for space reasons), results are generally weaker: For exact match, C-F1 values are about ∼10% points below those of $\mathtt{STag_{BLCC}}$, while approximate match performances are much closer. Hence, the independence assumptions of the BL tagger apparently lead to more 'local' errors. An analogous trend holds for argument relations.

*Additional Tasks:* We find that when we train $\mathtt{STag_{BL}}$ with only its main task—with label set $\mathcal{Y}$ as in Eq. (1)—the overall result is worst. In contrast, when we include the 'natural subtasks' "C" (label set $\mathcal{Y}_C$ consists of the projection on the coordinates $(b, t)$ in $\mathcal{Y}$) and/or "R" (label set $\mathcal{Y}_R$ consists of the projection on the coordinates $(d, s)$), performance increases typically by a few percentage points. This indicates that complex sequence tagging may benefit when we train a "sublabeler" in the same neural architecture, a finding that may be particularly relevant for morphological POS tagging (Müller et al., 2013). Unlike Søgaard and Goldberg (2016), we do not find that the optimal architecture is the one in which "lower" tasks (such as C or R) feed from lower layers. In fact, in one of the best parametrizations the C task and the full task feed from the same layer in the deep BiLSTM. Moreover, we find that the C task is consistently more helpful as an auxiliary task than the R task.

|  | C-F1 | | R-F1 | | F1 | |
|---|---|---|---|---|---|---|
|  | 100% | 50% | 100% | 50% | 100% | 50% |
| $\mathcal{Y}$-3 | 49.59 | 65.37 | 26.28 | 37.00 | 34.35 | 47.25 |
| $\mathcal{Y}$-3:$\mathcal{Y}_C$-1 | 54.71 | 66.84 | 28.44 | 37.35 | 37.40 | 47.92 |
| $\mathcal{Y}$-3:$\mathcal{Y}_R$-1 | 51.32 | 66.49 | 26.92 | 37.18 | 35.31 | 47.69 |
| $\mathcal{Y}$-3:$\mathcal{Y}_C$-3 | **54.58** | 67.66 | **30.22** | **40.30** | **38.90** | **50.51** |
| $\mathcal{Y}$-3:$\mathcal{Y}_R$-3 | 53.31 | 66.71 | 26.65 | 35.86 | 35.53 | 46.64 |
| $\mathcal{Y}$-3:$\mathcal{Y}_C$-1:$\mathcal{Y}_R$-2 | 52.95 | **67.84** | 27.90 | 39.71 | 36.54 | 50.09 |
| $\mathcal{Y}$-3:$\mathcal{Y}_C$-3:$\mathcal{Y}_R$-3 | 54.55 | 67.60 | 28.30 | 38.26 | 37.26 | 48.86 |

Table 2: Performance of MTL sequence tagging approaches, essay level. Tasks separated by ":". Layers from which tasks feed are indicated by respective numbers.

On essay level, **(d) LSTM-ER** performs very well on component identification (+5% C-F1 compared to $\mathtt{STag_{BLCC}}$), but rather poor on relation identification (-18% R-F1). Hence, its overall F1 on essay level is considerably below that of

STag$_{BLCC}$. In contrast, LSTM-ER trained and tested on paragraph level substantially outperforms all other systems discussed, both for component as well as for relation identification.

We think that its generally excellent performance on components is due to LSTM-ER's de-coupling of component and relation tasks. Our findings indicate that a similar result can be achieved for STag$_T$ via MTL when components and relations are included as auxiliary tasks, cf. Table 2. For example, the improvement of LSTM-ER over STag$_{BLCC}$, for C-F1, roughly matches the increase for STag$_{BL}$ when including components and relations separately ($\mathcal{Y}$-3:$\mathcal{Y}_C$-3:$\mathcal{Y}_R$-3) over not including them as auxiliary tasks ($\mathcal{Y}$-3). Lastly, the better performance of LSTM-ER over STag$_{BLCC}$ for relations on paragraph level appears to be a consequence of its better performance on components. E.g., when both arguments are correctly predicted, STag$_{BLCC}$ has even higher chance of getting their relation correct than LSTM-ER (95.34% vs. 94.17%).

Why does LSTM-ER degrade so much on essay level for R-F1? As said, text sequences are much longer on essay level than on paragraph level—hence, there are on average many more entities on essay level. Thus, there are also many more possible relations between all entities discovered in a text—namely, there are $O(2^{m^2})$ possible relations between $m$ discovered components. Due to its generality, LSTM-ER considers all these relations as plausible, while STag$_T$ does not (for any of choice of $T$): e.g., our coding explicitly constrains each premise to link to exactly *one* other component, rather than to $0, \ldots, m$ possible components, as LSTM-ER allows. In addition, our explicit coding of distance values $d$ biases the learner $T$ to reflect the distribution of distance values found in real essays—namely, that related components are typically close in terms of the number of components between them. In contrast, LSTM-ER only mildly prefers short-range dependencies over long-range dependencies, cf. Figure 4.

The **(e) ILP model** has access to both paragraph and essay level information and thus has always more information than all the neural systems compared to. Thus, it also knows in which paragraph in an essay it is. This is useful particularly for major claims, which always occur in first or last paragraphs in our data. Still, its performance is equal to or lower than that of LSTM-ER and STag$_{BLCC}$

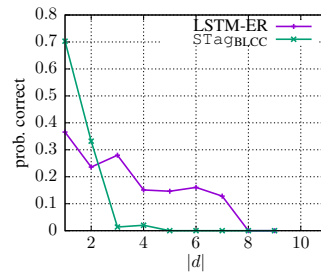

Figure 4: Probability of correct relation identification given true distance is $|d|$.

when both are evaluated on paragraph level.

# 6 Conclusion

We present the first study on neural end-to-end AM. We experimented with different *framings*, such as encoding AM as a dependency parsing problem, as a sequence tagging problem with particular label set, as a multi-task sequence tagging problem, and as a problem with both sequential and tree structure information. We show that (1) neural computational AM is as good or (substantially) better than a competing feature-based ILP formulation, while eliminating the need for manual feature engineering and costly ILP constraint designing. (2) BiLSTM taggers perform very well for component detection, as demonstrated for our STag$_T$ frameworks, for $T = $ BLCC and $T = $ BL, as well as for LSTM-ER (BLC tagger). (3) (Naively) coupling component and relation detection is not optimal, but both tasks should be treated seperately, but modeled jointly. (4) Relation detection is more difficult: when there are few entities in a text ("short documents"), a more general framework such as that provided in LSTM-ER performs reasonably well. When there are many entities ("long documents"), a more restrained modeling is preferable. These are also our *policy recommendations*. Our work yields new state-of-the-art results in end-to-end AM on the PE dataset from Stab and Gurevych (2016).

Another possible framing, not considered here, is to frame AM as an encoder-decoder problem (Bahdanau et al., 2015; Vinyals et al., 2015). This is an even more general modeling than LSTM-ER. Its suitability for the end-to-end learning task is scope for future work, but its adequacy for component typing and relation identification has been investigated in research submitted during the preparation of our current work (Potash et al., 2017).

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
