# Peer review of "Neural End-to-End Learning for Computational Argumentation Mining"

_ACL 2017 — decision unknown_

[Official Review · Reviewer 1 · rating 4 · confidence 4]
soundness 4 · originality 3 · clarity 5 · impact 2 · substance 3 · appropriateness 5 · meaningful comparison 3 · presentation format Poster

- Strengths:

The paper is well-written and easy to understand. The methods and results are
interesting.

- Weaknesses:

The evaluation and the obtained results might be problematic (see my comments
below).

- General Discussion:

This paper proposes a system for end-to-end argumentation mining using neural
networks. The authors model the problem using two approaches: (1) sequence
labeling (2) dependency parsing. The paper also includes the results of
experimenting with a multitask learning setting for the sequence labeling
approach. The paper clearly explains the motivation behind the proposed model.
Existing methods are based on ILP, manual feature engineering and manual design
of ILP constraints. However, the proposed model avoids such manual effort.
Moreover, the model jointly learns the subtasks in argumentation mining and
therefore, avoids the error back propagation problem in pipeline methods.
Except a few missing details (mentioned below), the methods are explained
clearly.

The experiments are substantial, the comparisons are performed properly, and
the results are interesting. My main concern about this paper is the small size
of the dataset and the large capacity of the used (Bi)LSTM-based recurrent
neural networks (BLC and BLCC). The dataset includes only around 320 essays for
training and 80 essays for testing. The size of the development set, however,
is not mentioned in the paper (and also the supplementary materials). This is
worrying because very few number of essays are left for training, which is a
crucial problem. The total number of tags in the training data is probably only
a few thousand. Compare it to the standard sequence labeling tasks, where
hundreds of thousands (sometimes millions) of tags are available. For this
reason, I am not sure if the model parameters are trained properly. The paper
also does not analyze the overfitting problem. It would be interesting to see
the training and development "loss" values during training (after each
parameter update or after each epoch). The authors have also provided some
information that can be seen as the evidence for overfitting: Line 622 "Our
explanation is that taggers are simpler local models, and thus need less
training data and are less prone to overfitting".

For the same reason, I am not sure if the models are stable enough. Mean and
standard deviation of multiple runs (different initializations of parameters)
need to be included. Statistical significance tests would also provide more
information about the stability of the models and the reliability of results.
Without these tests, it is hard to say if the better results are because of the
superiority of the proposed method or chance.

I understand that the neural networks used for modeling the tasks use their
regularization techniques. However, since the size of the dataset is too small,
the authors need to pay more attention to the regularization methods. The paper
does not mention regularization at all and the supplementary material only
mentions briefly about the regularization in LSTM-ER. This problem needs to be
addressed properly in the paper.

Instead of the current hyper-parameter optimization method (described in
supplementary materials) consider using Bayesian optimization methods.

Also move the information about pre-trained word embeddings and the error
analysis from the supplementary material to the paper. The extra one page
should be enough for this.

Please include some inter-annotator agreement scores. The paper describing the
dataset has some relevant information. This information would provide some
insight about the performance of the systems and the available room for
improvement.

Please consider illustrating figure 1 with different colors to make the quality
better for black and white prints.

Edit:

Thanks for answering my questions. I have increased the recommendation score to
4. Please do include the F1-score ranges in your paper and also report mean and
variance of different settings. I am still concerned about the model stability.
For example, the large variance of Kiperwasser setting needs to be analyzed
properly. Even the F1 changes in the range [0.56, 0.61] is relatively large.
Including these score ranges in your paper helps replicating your work.

[Official Review · Reviewer 2 · rating 4 · confidence 4]
soundness 4 · originality 3 · clarity 3 · impact 2 · substance 4 · appropriateness 5 · meaningful comparison 3 · presentation format Oral Presentation

The work describes a joint neural approach to argumentation mining. There are
several approaches explored including:
 1) casting the problem as a dependency parsing problem (trying several
different parsers)
 2) casting the problem as a sequence labeling problem
3) multi task learning (based on sequence labeling model underneath)
4) an out of the box neural model for labeling entities and relations (LSTM-ER)
5) ILP based state-of-the art models
All the approaches are evaluated using F1 defined on concepts and relations. 
Dependency based solutions do not work well, seq. labeling solutions are
effective.
The out-of-the-box LSTM-ER model performs very well. Especially on paragraph
level.
The Seq. labeling and LSTM-ER models both outperform the ILP approach.
A very comprehensive supplement was given, with all the technicalities of
training
the models, optimizing hyper-parameters etc.
It was also shown that sequence labeling models can be greatly improved by the
multitask
approach (with the claim task helping more than the relation task).
The aper  is a very thorough investigation of neural based approaches to
end-to-end argumentation mining.

- Major remarks  
  - my one concern is with the data set, i'm wondering if it's a problem that
essays in the train set and in the test set might
   be on the same topics, consequently writers might use the same or similar
arguments in both essays, leading to information
   leakage from the train to the test set. In turn, this might give overly
optimistic performance estimates. Though, i think the same
   issues are present for the ILP models, so your model does not have an unfair
advantage. Still, this may be something to discuss.

  - my other concern is that one of your best models LSTM-ER is acutally just a
an out-of-the box application of a model from related
    work. However, given the relative success of sequence based models and all
the experiments and useful lessons learned, I think this 
    work deserves to be published.

- Minor remarks and questions:
222 - 226 - i guess you are arguing that it's possible to reconstruct the full
graph once you get a tree as output? Still, this part is not quite clear.
443-444 The ordering in this section is seq. tagging -> dependency based -> MTL
using seq. tagging, it would be much easier to follow if the order of the first
two were
                  reversed (by the time I got here i'd forgotten what STag_T
stood for)
455 - What does it mean that it de-couples them but jointly models them (isn't
coupling them required to jointly model them?)
         - i checked Miwa and Bansal and I couldn't find it
477 - 479 -  It's confusing when you say your system de-couples relation info
from entity info, my best guess is that you mean it
                        learns some tasks as "the edges of the tree" and some
other tasks as "the labels on those edges", thus decoupling them. 
                        In any case,  I recommend you make this part clearer

Are the F1 scores in the paragraph and essay settings comparable? In particular
for the relation tasks. I'm wondering if paragraph based 
models might miss some cross paragraph relations by default, because they will
never consider them.